Review

# Exploring the basis of heterogeneity of cancer aggressiveness among the mutated POLE variants

Janick Selves[1,2], Helena de Castro e Gloria[3], Anne-Cécile Brunac[1], Jenifer Saffi[3], Rosine Guimbaud[2,4,5], Pierre Brousset[1,2,6], Jean-Sébastien Hoffmann[1,6]

Germline pathogenic variants in the exonuclease domain of the replicative DNA polymerase Pol ε encoded by the *POLE* gene, predispose essentially to colorectal and endometrial tumors by inducing an ultramutator phenotype. It is still unclear whether all the *POLE* alterations influence similar strength tumorigenesis, immune microenvironment, and treatment response. In this review, we summarize the current understanding of the mechanisms and consequences of *POLE* mutations in human malignancies; we highlight the heterogeneity of mutation rate and cancer aggressiveness among POLE variants, propose some mechanistic basis underlining such heterogeneity, and discuss novel considerations for the choice and efficacy of therapies of POLE tumors.

## Introduction

### Importance of the 3′–5′ exonuclease proofreading activity of DNA polymerase ε for accurate duplication of genomic DNA in human cells

In human cells, the accurate duplication of the six billion nucleotides that constitute the genome requires the action of the processive "replicative" DNA polymerases Pols δ and ε, also named "high-fidelity" or "error-free" polymerases, the main actors at the replication forks (see Fig 1A) (Miyabe et al, 2011). It is now widely accepted that Pol δ replicates the lagging strand, whereas Pol ε replicates the leading strand (Baris et al, 2022). The high accurate action of these enzymes relies on three major processes: (i) the strong intrinsic base selectivity of Pol δ and Pol ε; (ii) the presence of an associated 3′–5′ exonuclease proofreading activity (Exo), which detects mismatches, insertions, and deletions and is able to switch the 3′ end of the incorrectly extended DNA strand to an exonuclease active site, where the aberrant bases can be removed;

and (iii) a post-replicative mismatch repair (MMR) mechanism performed by protein complexes comprising MutL homolog 1 (MLH1), MutS protein homolog 2 (MSH2), MutS homolog 6 (MSH6), and PMS1 homolog 2 (PMS2) (Burgers & Kunkel, 2017). Both Exo and MMR mechanisms maintain genome stability by correcting mis-incorporated non-complementary nucleotides, and compromising one of these activities can lead to cancer syndromes characterized by an increased rate of replicative mutagenesis.

Studies in yeast have unveiled that the properties of Pol ε differ from those of Pol δ. First, Pol ε has a high-molecular-weight subunit whose N-terminal domain encodes DNA polymerization and 3′ exonuclease activity, and a catalytically inactive C-terminal domain that is required for replisome assembly and checkpoint activation. Second, Pol ε has a small domain in the catalytic subunit allowing it to encircle the nascent dsDNA and conferring a high intrinsic processivity, the number of nucleotidyl transfer reactions occurring during a single Pol–DNA binding event (Hogg et al, 2014). Third, Pol ε possesses a particularly efficient intrinsic 3′ exonuclease activity for its own replication errors, which impedes an efficient strand displacement synthesis activity and renders Pol ε more accurate than Pol δ.

Among the four subunits that constitute human Pol ε, the largest one is encoded by *POLE* and comprises both the catalytic and proofreading exonuclease activities (Shevelev & Hubscher, 2002). The other subunit encoded by *POLE2* facilitates the interaction with GINS and targets Pol ε to the leading strand during the initiation of DNA replication (Langston et al, 2014), whereas *POLE3* and *POLE4* are important for the binding to the double-stranded DNA, processive DNA synthesis, and processive 3′–5′ exonuclease degradation (Aksenova et al, 2010). Several highly conserved catalytic residues D275 and E277 in the *POLE* exonuclease domains are required for the proofreading function of the polymerase. Because of its reduced efficiency in extending a mispaired primer terminus, Pol ε pauses when a nucleotide is misincorporated leading to a switch from the catalytic to the exonuclease domain where the incorrect nucleotide is excised and the correct one is inserted before DNA synthesis

[1]Department of Pathology, Institut Universitaire du Cancer-Oncopole de Toulouse; Centre Hospitalier Universitaire (CHU), Toulouse, France  [2]Université Fédérale Toulouse Midi-Pyrénées, Université Toulouse III Paul Sabatier, INSERM, CRCT, Toulouse, France  [3]Laboratory of Genetic Toxicology, Federal University of Health Sciences of Porto Alegre, Porto Alegre, Brazil  [4]Department of Digestive Oncology, Centre Hospitalier Universitaire (CHU), Toulouse, France  [5]Department of Digestive Surgery, Centre Hospitalier Universitaire (CHU), Toulouse, France  [6]Laboratoire d'Excellence Toulouse Cancer (TOUCAN), Toulouse, France

Correspondence: jean-sebastien.hoffmann@inserm.fr; selves.j@chu-toulouse.fr

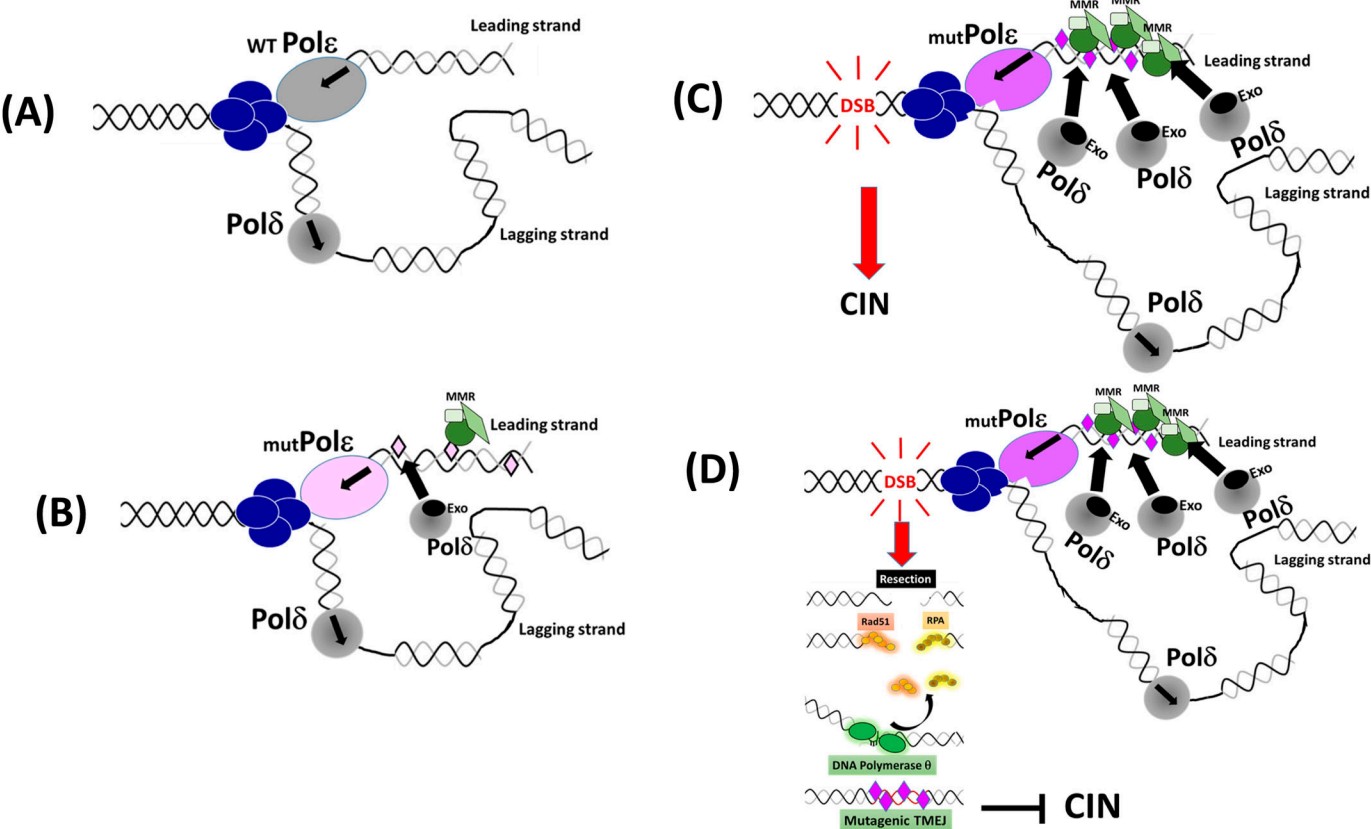

**Figure 1.   Potential impact of different mutated Pole at the replication fork.**
**(A)** Representative DNA replication fork in human cells where Pol ε acts as the leading strand polymerase. In blue, the helicase complex opening the duplex DNA.
**(B)** Recent studies using the yeast model have demonstrated that despite the important change in structural and biochemical properties, mutated Pol ε associated with human cancers remains leading strand replicates and generates a high level of errors on this strand (Bulock et al, 2020). It was also documented that both MMR and extrinsic proofreading by Pol δ operate to correct some mutated Pol ε errors (Bulock et al, 2020), so the resulting error rate could be compatible with cell survival and sufficient to cause a tumor. This mechanism might concern an important subpopulation of *POLE* mutants. **(C)** When the error rate is very high by an ultramutator Pol ε, the strong need for MMR complexes and Pol δ Exo to access the mismatches could lead to a slow-down progression of Pol ε and frequent Pol ε dissociation from the primer termini; in turn, this could interfere with the coordination of the leading and lagging strand synthesis and affect the global progression of the fork. Persisting fork stalling could lead then to fork collapse and chromosomal breakage, further enhancing manifestations of Chromosomal Instability (CIN) in these tumors. **(D)** In the case of excessive DNA damage, the alternative TMEJ may operate to limit the accumulation of excessive DNA breakage and CIN (Maiorano et al, 2021). TMEJ operates preferentially on resected double-strand breaks (DSB), Pol θ is recruited and stabilizes the resected DSB synapse, RPA and RAD51 are removed, and Pol θ detects internal microhomologies that can be annealed. Pol θ then starts DNA synthesis with low processivity and numerous aborted syntheses resulting in a high rate of deletions and insertions (Prodhomme et al, 2021). These ultra-hypermutated cells mutate continuously, potentially generating multiple independent subclones, and such a high mutation load may represent an Achilles' heel, exploitable for therapeutic intervention.

restarts (Ganai et al, 2015). This editing function seems to enhance the fidelity by about 200-fold.

The balance between exonuclease and polymerase activities allows DNA synthesis over degradation when nucleotides are correctly added, but incorporation of a wrong nucleotide shifts such equilibrium toward the exonuclease site. After edition of the incorrect nucleotides, the balance is again in favor of DNA synthesis. Thus far, several structures of family B DNA polymerases with proofreading activity from different species have been determined, and their overall structures comprising a thumb, palm, fingers, and the exonuclease domain are well conserved. These structures provided insight into the mechanism of proofreading activity and have served as a paradigm for understanding the editing process of the human replicative polymerases. It appears that the polymerase and the exonuclease sites of Pol ε are positioned in separate domains away

from each other of about 35–40 Å. Such a distance obliges three to four nucleotides of the nascent DNA strand to be unpaired and induces a major structural change for switching from polymerization to proofreading. Such a change involves a transfer of the 3'-terminus of the DNA via an intramolecular mechanism (Hogg et al, 2014; Ganai et al, 2015). The exonuclease domain of the replicative polymerases shares a common architecture containing exonuclease (Exo) I–III motifs, which are highly conserved in A and B DNA polymerase families from viral, prokaryotic, and eukaryotic origins, and found in the superfamily of nucleases that includes RNases and DNases.

### *POLE* mutations and cancer

The best-characterized mutator phenotype in cancer involves germline mutations in MMR genes for patients with Lynch syndrome, which

causes predisposition to colorectal cancer (CRC) and an increased risk of several other cancer types. Such syndrome is caused by germline heterozygous inactivating mutations in one of the MMR genes followed by somatic loss of the remaining allele, leading to MMR deficiency (Peltomaki, 2003). Somatic MMR defects are also common in sporadic digestive cancers, notably through the transcriptional silencing by methylation of the *MLH1* promoter, leading to microsatellite instability (MSI) (Kane et al, 1997; Vilar & Gruber, 2010). MSI is most prevalent in endometrium (30%), stomach (20%), and colon and rectum cancers (15%) including the Lynch syndrome patients, and occurrences of MSI are rarely observed in other cancer types. Frequent insertion and deletion mutations (indels), especially at homopolymer and dinucleotide repeats, as well as a global mutator phenotype (10–100 mutations per Mb) and reduced chromosomal instability, define these MSI tumors, in contrast to microsatellite stable tumors of the same type, which exhibit a much reduced mutation frequency (1–10 mutations per Mb) and increased chromosomal instability (Ciriello et al, 2013).

The association between replicative DNA polymerase defects and cancer in humans is more recent, notably Pol ε, which has the highest accuracy among DNA polymerases. The last decade has witnessed the identification in cancers from many tissue types of multiple somatically acquired missense mutations clustering in the sequence encoding the exonuclease proofreading domain of *POLE* with a high incidence in colorectal (3%) and endometrial (8%) cancers (Yoshida et al, 2011; Cancer Genome Atlas Research Network et al, 2013; Palles et al, 2013; Heitzer & Tomlinson, 2014; Shinbrot et al, 2014; Campbell et al, 2017). The fact that *POLE* mutations found in hypermutated tumors are located within highly conserved amino acid residues in the exonuclease domain strongly supports that loss of proofreading activity is responsible for the accumulation of mutations in these cancers. Such a defective proofreading activity producing a mutator phenotype, which has been established in model systems, such as yeast, bacteria, and mice, leads to tumorigenesis. These *POLE* variants are present in heterozygous tumors with no apparent loss of heterozygosity and with high mutation loads, up to 500 mutations per megabase (Mut/Mb) (Palles et al, 2013). Such mutator phenotype in the presence of the WT allele is consistent with the participation of both the WT and the mutant polymerases in DNA replication, in contrast to mutated MMR tumors, where the loss of both alleles is required to produce a mutator effect. It has been proposed that differential expression levels of the WT and mutant *POLE* alleles in the course of cancer progression may allow transient stages of hypermutation that promote tumor growth together with a threshold limiting excessive mutation load to maintain fitness (Daee et al, 2010). *POLE* mutations have also been detected, albeit less frequently, in other types of gastrointestinal cancer, and in brain, breast, ovary, prostate, lung, kidney, cervix, and bone tumors (Cancer Genome Atlas Network, 2012; Cerami et al, 2012; Cancer Genome Atlas Research Network et al, 2013; Shinbrot et al, 2014; Forbes et al, 2015; Grossman et al, 2016; Campbell et al, 2017).

Mutations in the exonuclease domain of POLE can also be inherited through the germline, leading to a rare autosomal dominant familial cancer predisposition syndrome documented as polymerase proofreading–associated polyposis (PPAP), characterized primarily by early-onset colorectal and endometrial tumors (see Table 1). The PPAP phenotype overlaps with that of Lynch syndrome and MUTYH-associated polyposis, and screening and management algorithms are broadly similar. A recent comprehensive description of the clinicopathological features of PPAP reported that the cumulative incidence of CRC in POLE variant heterozygotes is almost 90% (Palles et al, 2022). Most *POLE* patients have a reduced polyposis phenotype (10–100 polyps), and a minority of cases have a Lynch syndrome–like phenotype with early-onset CRC and few polyps. Endometrial and esophageal cancers are the most frequent extraintestinal malignancies with lower risks than CRC, and there is growing evidence that a variety of brain tumors can occur (Vande Perre et al, 2019). These germline *POLE* mutations include the highly penetrant *POLE*-L424V variant, which predisposes to multiple colorectal adenomas and carcinomas (Palles et al, 2013). A recent report has evidenced that cancer-associated *POLE* alleles can lead to a mutator phenotype even when MMR is functional. In these tumors, insertions are predominantly found, confirming that the proofreading activity repairs extra bases on the nascent strand by shifting this strand into the exonuclease domain where it can be repaired, and that mutations in the proofreading domain would prevent this shift and lead to permanent insertions as replication continues (Chung et al, 2021). This is different than the MMR system, which is more effective in detecting loops on the parental strand, these loops resulting in deletions if unrepaired in MMR-defective cancers.

Several *POLE* variants within the exonuclease domain such as P286R, V411L, L424V, S459F, P286H, F367S, and L424I show a decrease in exonuclease activity, as measured by biochemistry experiments using purified Pol ε (Korona et al, 2011; Shinbrot et al, 2014; Parkash et al, 2019).

### Heterogeneity of the POLE mutation impact

To distinguish between *POLE* mutations that are responsible for driving a mutator phenotype from *POLE* mutations that are nonpathogenic or passengers, it has been proposed that *POLE* mutation drivers can be defined by their occurrence in hypermutated tumors. Generally, there are much less cancer driver mutations in *POLD1* than in *POLE* in human cancers. This might be due to the reduced fitness and viability of *POLD1* mutants as Pol δ holds multiple critical roles besides lagging strand replication, including its ability to proofread in trans the errors made by Pol ε and Pol α, and its role during MMR and during Okazaki fragment maturation. Interestingly, the severity of the proofreading and fidelity defects observed for *POLE* variants is not correlated to their frequency detected in cancers, implying that besides proofreading loss, additional mechanisms drive tumor initiation and progression. Some variants are even associated with a higher mutation rate compared to Pol ε lacking the proofreading domain (Kane & Shcherbakova, 2014). Recent works started to explore a mechanistic explanation and unveiled that the P286R mutations, by preventing the access of the 3′-terminus to the exonuclease site during elongation, favor the annealing at the polymerase site and lead to a relatively more active DNA polymerization activity, which can extend mispaired termini or progress through non-B structured DNA with higher efficiency. As a consequence, such a hyperactive polymerization aptitude has a strong mutagenic impact on the replicative forks,

**Table 1. Cancer-associated POLE variants.**

| POLE mutation | Mutation domain | Tumor type | Mutation in yeast | Fold increase in Mutation Frequency in *S. cerevisiae* | Reference |
|---|---|---|---|---|---|
| D275A | Exo I | CRC | n.a | n.a | Yoshida et al (2011) |
| T278L | Exo I | CRC, EC | pol2-T279L | n.a | Cerami et al (2012), Gao et al (2013), and Castellsague et al (2019) |
| P286R | Exo I | CRC, EC, BC, GBM, EOC, PC | pol2-P301R | 27 | Cerami et al (2012), Cancer Genome Atlas Research Network et al (2013), Gao et al (2013), Kane & Shcherbakova (2014), Erson-Omay et al (2015), Hoang et al (2015), Campbell et al (2017), and Maruvka et al (2017) |
| S297F | between Exo I and Exo II | EC, GBM, CC, EOC | n.a | n.a | Cerami et al (2012), Cancer Genome Atlas Research Network et al (2013), Gao et al (2013), and Zou et al (2014) |
| F367S | Exo II | CRC | pol2-F382S | 17 | Yoshida et al (2011), Cerami et al (2012), Gao et al (2013), and Barbari et al (2018) |
| V411L | between Exo II and Exo III | CRC, EC, GBM, GC, EOC, CC, GC | po12-V426L | 1.2 | Cancer Genome Atlas Network (2012), Cerami et al (2012), Cancer Genome Atlas Research Network et al (2013), Gao et al (2013), Palles et al (2013), Hoang et al (2015), Campbell et al (2017), Maruvka et al (2017), and Barbari et al (2018) |
| L424V | between Exo II and Exo III | CRC, EC, BC | pol2-L439V | 5.2 | Cerami et al (2012), Gao et al (2013), Palles et al (2013), Campbell et al (2017), and Barbari et al (2018) |
| S459F | Exo III | CRC, EC, AA | pol2-5474F | 30 | Cerami et al (2012), Gao et al (2013), Shlien et al (2015), and Barbari et al (2018) |
| 5461P | Exo III | EC, PN | n.a | n.a | Cerami et al (2012), Gao et al (2013), Shlien et al (2015), and Campbell et al (2017) |
| P436R | Exo V | CRC, EC | pol2-P451R | 5.2 | Cancer Genome Atlas Network (2012), Billingsley et al (2015), and Barbari et al (2018) |
| M444K | Exo V | EC | pol2-M459K | n.a | Cancer Genome Atlas Research Network et al (2013) |
| A456P | Exo III | CRC, EC | n.a | n.a | Cancer Genome Atlas Research Network et al (2013), Church et al (2013), Shinbrot et al (2014), Stenzinger et al (2014), and Billingsley et al (2015) |
| N363K | Exo II | CRC, GBM | pol2-N378K | 43 | Vande Perre et al (2019), Dahl et al (2022), and Labrousse et al (2023) |

CRC, colorectal cancer; EC, endometrial cancer; GBM, glioblastoma; AA, anaplastic astrocytoma; PN, primitive neuroectodermal tumor; BC, breast cancer; SCCC, squamous cell cervical carcinoma; CC, cervical carcinoma; EOC, endometrioid ovarian carcinoma; GC, gastric cancer.

stronger than the influence of a Pol ε devoid of exonuclease (Parkash et al, 2019; Xing et al, 2019). Whether other *POLE* cancer mutant alleles could drive tumor progression in a similar manner is an important issue for future works. Recent data obtained in yeast unveiled that the hypermutator phenotype associated with mutations in the DNA binding cleft of the exonuclease domain in Pol ε may be achieved not only by changes in the balance between its DNA synthesis and proofreading abilities but also by its enhanced processivity, decreasing the probability for extrinsic proofreading (Dahl et al, 2022).

Modelization in yeast of these *POLE* mutations has allowed the study of their mutagenic impact and unveiled that the mutation rate can vary by more than two orders of magnitude depending on the *POLE* variant (Daee et al, 2010; Kane & Shcherbakova, 2014; Mertz et al, 2015; Esteban-Jurado et al, 2017; Barbari et al, 2018, 2022; Parkash et al, 2019; Soriano et al, 2021). Surprisingly, these differences in mutagenesis do not reflect the frequency nor the level of Tumor mutational burden (TMB). For example, P286R and V411L, which are the two most frequent *POLE* mutations in cancer inducing high TMB, show very different impacts of the mutation

rate in haploid and diploid yeast. The P286R mutant induces an increase in the mutation rate by 150-fold, whereas the V411L mutant does not show any difference as compared to the WT. In addition, very modest mutator alleles can cause synergistic increases when combined with defects in MMR.

Besides the DNA mismatch repair defects that underlie Lynch syndrome, the mutations in *POLE* highlight the critical role of replication errors in predisposition to colorectal and endometrial cancers. This is in contrast to cancers of the breast and ovary, in which double-stranded DNA break repair is more significant in predisposition (Heeke et al, 2018).

## Mutator phenotype and mutation signature in POLE variants

Generally, cancers exhibiting a high level of nucleotidic mutations contain clonal driver mutations present in the vast majority of the tumor cells, which are positively selected and drive malignant progression. They also contain subclonal mutations randomly distributed through the genome, providing a vast reservoir of variant cells contributing to tumor heterogeneity, some of which

being able to be selected when exposed to anti-cancer treatment, expand, and lead to the emergence of therapeutic resistance (Loeb et al, 2019). This is particularly true for mutated *POLE* cancers, where the strong accumulation of mutations can provide such a reservoir. Although the proofreading activity of the lagging strand replicative polymerase Pol $\delta$ can also act on the leading strand and correct excessive errors generated by mutated Pol $\varepsilon$ (Fig 1) (Bulock et al, 2020), a high increase in mutation rates has been documented in cancers with P286R, D275V, P286H, F367S, L424V, P436R, and S459F changes located closed to the DNA binding cleft of Pol $\varepsilon$ (Cancer Genome Atlas Network, 2012; Cancer Genome Atlas Research Network et al, 2013; Kane & Shcherbakova, 2014; Shinbrot et al, 2014; Barbari et al, 2018). The scale of these mutator phenotypes is highly variable and seems to be connected to the incidence of these variants in tumors (Barbari et al, 2018). Importantly, all of these variants lead to a much higher mutation frequency as compared to the variant that completely eliminates proofreading.

*POLE* tumors can be stratified into distinct groups based on the relative abundance of *POLE* and *POLE*/MMR mutation signatures. Among hypermutated cancers presenting ≥10 Mut/Mb, ~25% are associated with mutations in MMR genes alone, whereas a large number of ultra-hypermutated cancers (≥100 Mut/Mb) show mutations affecting both MMR and the replicative DNA polymerases, mainly *POLE* (Jansen et al, 2016). Some tumors with biallelic MMR deficiency and acquired somatic heterozygous mutations in *POLE* accumulate one of the highest mutation burdens observed thus far and expand very aggressively (Shlien et al, 2015; Campbell et al, 2017). They include Lynch syndrome tumors (Jansen et al, 2016), as well as endometrial, colorectal cancers and glioblastoma (Shlien et al, 2015).

Specific signature errors have been shown to be enriched in mutated *POLE* cancers. These include three trinucleotide hotspot mutations: C>A transversions in the TCT context (C>A-TCT), C>T-TCG, and T>G-TTT. Complex mutation signature analyses have divided these into the three distinct single-base substitution signatures: 10a (C>A-TCT), 10b (C>T-TCG), and 28 (T>G-TTT) (Cancer Genome Atlas Network, 2012; Alexandrov et al, 2013; Cancer Genome Atlas Research Network et al, 2013; Alexandrov & Stratton, 2014; Shinbrot et al, 2014; Campbell et al, 2017; Haradhvala et al, 2018), as well as double-base substitution DBS3 signature (Alexandrov et al, 2020). The two most frequent *POLE* mutations, P286R and V411L, have different compatibilities with MMR and drive different amounts of 10a and 10b signature mutations. The signature single-base substitution 14, which is largely composed of C>A transversions in NCT trinucleotide contexts, has been identified in mutated *POLE* tumors inactivated for MMR (Haradhvala et al, 2018). A recent work demonstrated that Pol $\varepsilon$ mutants can, by themselves, drive an ultra-hypermutated phenotype independent of MMR inactivation and MSI (Hodel et al, 2020). The mutation spectrum in these tumors results from the specific mutant allele, its abundance in the cell, and the status of MMR (Hodel et al, 2020).

It was recently reported that besides tumor DNA, increased mutation loads were also found in normal tissue from individuals with germline *POLE* mutations although they do not display obvious signs of premature aging (Robinson et al, 2021). This implies not only that human physiology can tolerate ubiquitously elevated mutation burdens even during development and that aging is not entirely attributable to the malfunction process induced by somatic

mutations accumulated during life, but also that cancer risk does not simply rely on increased mutation rate.

## Mutated *POLE* tumors and immunogenic response

It is becoming increasingly clear that cancers with a high intrinsic mutation load (MMR- and *POLE*-mutated cancers) or cancers related to mutagenic environmental genotoxic exposure such as lung, melanoma, and bladder cancers, which show all a high mutation burden, respond generally well to immunotherapy (Rizvi et al, 2015; McGranahan et al, 2016; Yarchoan et al, 2017). The MMR-deficient CRC tumors have a remarkably favorable prognosis despite their early onset and rapid progression and respond well to immune checkpoint blockades such as $\alpha$-PD-1, $\alpha$-PD-L1, and $\alpha$-CTLA-4, suggesting that their intrinsic high mutation load could trigger chronic immune surveillance that can be further enhanced by immunotherapy (Le et al, 2015, 2017). In the case of the mutated *POLE* tumors, the high level of nucleotide insertions inside the genomic DNA produces highly immunogenic neoantigens, which in turn recruits tumor-infiltrating lymphocytes (TILs) resulting in a strong immunogenic response against these tumors (Keshinro et al, 2021). Accordingly, mutated *POLE* tumors are enriched in TILs for endometrial, colorectal, lung, and brain cancers and measuring the abundance of these TILs might be a better predictor of prognosis than MSI status. Patients with hypermutated endometrial and CRC with mutated *POLE* have an excellent prognosis with a very high percentage of progression-free survival after surgery (van Gool et al, 2016). This might imply that hypermutated *POLE* tumors, usually treated with radiation and chemotherapy, could be handled less aggressively despite their higher grade and that immunotherapy alone might be sufficient. The excellent prognosis of mutated *POLE*/CRC seems to be better than mutated MMR/CRC, supporting that monitoring *POLE* mutation within clinical testing panels could improve risk stratification in somatic CRC. *POLE* is well appropriate to NGS panels because its exonuclease domain is relatively small, its mutations are clustering at recurrent hotspots, and germline *POLE* variants might be easily detected (Domingo et al, 2016).

It was recently demonstrated that *POLE* mutations affecting only proofreading could predict anti–PD-1 efficacy in mismatch repair–proficient tumors revealing *POLE* proofreading deficiency as a new tissue-agnostic biomarker for cancer immunotherapy (Rousseau et al, 2022).

However, the correlation between the mutation load and response to immunotherapy is not absolute, as many hypermutated tumors do not respond to the immune checkpoint blockade. Ongoing and future works need to shed light on these issues and identify optimal biomarkers of sensitivity to the immune checkpoint blockade. One of the problems relies on the assays used to assess TMB, whose estimation is easily prone to bias, so the determination of optimal threshold to predict response to immunotherapy should be improved and standardized. The predictive ability of mutagenic load may be also further enhanced by the use of additional biomarkers. Another possible explanation comes from the idea that not all types of mutation are equally efficient in generating neoantigens. This has been documented for non–small-cell lung cancer, where non-synonymous mutation burden correlated better with progression-free survival upon the PD-1 blockade compared to the total exonic mutation landscape (Rizvi

et al, 2015). Moreover, CRC tumors with large numbers of frameshift mutations, which are believed to better generate neoantigens, show higher numbers of CD+ T-cell infiltrates (Maby et al, 2015). One additional explanation that we will discuss more in deep in the next paragraph is the ability of a variant, besides a mutator phenotype, to create DNA damage and accumulate numerical and structural changes, referred to as chromosomal instability (CIN).

### Replication stress, DNA damage, and CIN in POLE variant

Many solid cancers and some hematological malignancies are characterized by CIN, which has been extensively linked to tumorigenesis, cancer progression, and tumor resistance (Hanahan & Weinberg, 2000). The link between CIN and replication problems has been largely documented. When the replication forks encounter endogenous DNA distortions within repetitive sequences, non-B structured DNA, persistent base alterations induced by external chemical or physical carcinogens, or even transcriptional machinery, they frequently halt, a process referred to as replicative stress (RS) that strongly affects genome stability (Zeman & Cimprich, 2014). Indeed, failure to stabilize and restart stalled forks or prolonged arrest of replication forks may result in fork collapse, leading to chromosomal breakage such as DSBs and CIN. In general, there is no overlapping between the tumors that express a hypermutated phenotype and the tumors that are characterized by CIN. Many POLE/Exo-mutated cancers described thus far display a high mutator phenotype without affecting chromosomal integrity (see the model in Fig 1B). Recently, we have reported an exceptional cancer-related hereditary POLE mutation, N363K, which not only affects proofreading and exhibits a high degree of inaccurate DNA synthesis, but also induces chromosomal breakage, chromosomal aberrations, and aneuploidy (Labrousse et al, 2023). A hypothetic mechanistic model that may explain such features is given in Fig 1C. A high rate of misincorporation might recruit abundant MMR actors and proofreading domains from lagging strand replicative Pol $\delta$, which in turn might create a strong slow-down of Pol $\varepsilon$, as well as uncoupling leading versus lagging strand, a situation that can arrest the progression of the fork and increase the probability of fork collapse and chromosomal breakage. This model might explain why this variant produces giant cell glioblastoma, a rare cancer characterized by extremely high DNA damage (Rohlin et al, 2014; Vande Perre et al, 2019) in addition to the typical colon and endometrial tumor spectrum (Labrousse et al, 2023). These findings may influence the therapeutic strategies. Indeed, aneuploid tumors could be targeted by inhibiting crucial actors involved in the DNA damage responses, such as the checkpoint kinases ATR, CHK1, and WEE1. Inhibiting these kinases generates DNA fragments into the cytoplasm and in turn stimulates an innate immune response, so patients with CIN POLE tumors could benefit from the combined action of DNA damage response inhibitors and immune checkpoint inhibitors. Furthermore, excessive DNA breaks induced by fork collapse in some POLE variants might be repaired by the alternative mutagenic Pol theta-mediated end-joining pathway and therefore be enriched in targeted mutations by the central actor of theta-mediated end-joining, the error-prone DNA polymerase Pol $\theta$ (see the model in Fig 1D). Such a process is believed to limit lethal excessive chromosomal abnormalities while still driving tumor cell diversification and heterogeneity upon which selection and Darwinian evolution can still act (Maiorano et al, 2021).

Therefore, this category of cancers with DNA damage supplemented by targeted repair mutagenesis could respond to immunotherapy.

## Conclusion

Depending on the level of CIN versus mutator phenotype, POLE tumors could present various degrees of aggressiveness and differential responses to immunotherapy. Such exploration and knowledge for the minority of POLE-mutated cancers that escape immunotherapy might help for the orientation of alternative therapies of patients and implementation of surveillance to their family members. Whether replication checkpoint kinases or alternative DSB repair actors could be Achilles' heels of such rare CIN POLE tumors needs to be considered in the near future.

## Supplementary Information

### Author Contributions

J Selves: conceptualization and writing—review and editing.
H de Castro e Gloria: conceptualization and writing—original draft.
A-C Brunac: conceptualization.
J Saffi: conceptualization.
R Guimbaud: conceptualization.
P Brousset: conceptualization.
J-S Hoffmann: conceptualization and writing—original draft, review, and editing.

### Conflict of Interest Statement

The authors declare that they have no conflict of interest.

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
