## [Reviewer comments · Life Science Alliance]

Life Science Alliance

Exploring the basis of heterogeneity of cancer aggressiveness among the mutated POLE variants

Janick Selves, Helena DE CASTRO E GLORIA, Anne-Cécile BRUNAC, Jenifer SAFFI, Rosine GUIMBAUD, Pierre Brousset, and Jean-Sébastien Hoffmann

DOI: <https://doi.org/10.26508/lsa.202302290>

Corresponding author(s): Jean-Sébastien Hoffmann, University Cancer Institute Toulouse Oncopole and Janick Selves, University Cancer Institute Toulouse Oncopole

Review Timeline:

Submission Date:	2023-07-25
Editorial Decision:	2023-08-31
Revision Received:	2023-09-28
Editorial Decision:	2023-10-02
Revision Received:	2023-10-04
Accepted:	2023-10-18

Transaction Report:

August 31, 2023

Re: Life Science Alliance manuscript #LSA-2023-02290-T

Dr. Jean-Sébastien Hoffmann
INSERM
CRCT
2, Avenue Hubert Curien
IUCTO
Toulouse 31037
France

Dear Dr. Hoffmann,

Thank you for submitting your manuscript entitled "Exploring the basis of heterogeneity of cancer aggressiveness among the mutated POLE variants" to Life Science Alliance. The manuscript was assessed by expert reviewers, whose comments are appended to this letter. We invite you to submit a revised manuscript addressing the Reviewer comments.

Thank you for this interesting contribution to Life Science Alliance. We are looking forward to receiving your revised manuscript.

Sincerely,

B. MANUSCRIPT ORGANIZATION AND FORMATTING:

Reviewer #1 (Comments to the Authors (Required)):

This is a review paper which attempts to describe the current knowledge about POLE variants, their influence on tumor mutation rates and clinical implications of POLE mutations. Overall it does cover a lot of the material that is currently in the literature about POLE variants. However, there are other recent reviews including (PMID: 33255191, PMID: 37388540) that describe more detail about the points raised in this review. The current manuscript could add to the field by exploring some of the issues not discussed in other reviews such as comparing the clinical relevance of different variants, a direct comparison between germline and somatic variants, response to therapeutics in POLE mutant cancers, morphological differences that are found in POLE mutant endometrial and colorectal cancers, and variants of unknown significance and how to determine their impact, as well as when the POLE mutations are thought to occur relative to other cancer driver mutations. The comparison of POLE and POLD1 mutations in hypermutated cancers is also something that could be added to make this review more robust.

Overall, in many sections, primary and recent references are missing or incomplete. There are statements throughout the manuscript that warrant an original citation where none is added, or some sentences list a few recent citations while missing the original papers. For a review paper, the citations are a critical component. These authors should provide more comprehensive references and citations. It is expected that a review will include both original and recent citations, and present a comprehensive assessment of the literature. The inconsistent and incomplete references is the major shortcoming of this manuscript.

Importance of the 3' to 5' exonuclease proofreading activity of DNA Polymerase for accurate duplication of genomic DNA in human cells

This section compromises a decent evaluation of the current understanding of the mechanism of action of POLE.

POLE mutations and cancer

1. Major Criticism: This section starts with a description of MMR deficient cancers, then progresses into a description of POLE deficient cancers. The references for many sentences are incomplete or missing, here are some examples, (page 5) "POLE mutations have also been detected, albeit less frequently, in other types of gastrointestinal cancer, as well as in brain, breast, ovary, prostate, lung, kidney, cervix, and bone tumors (13,20-22)." does not include earlier papers that initially describe the variants, such as (17,59,60) and does reference databases such as COSMIC, cbiportal which is a bit confusing. Not including the appropriate references is a major problem throughout this manuscript.

2. The end of this section (page6) contains this sentence, "Several POLE variants within the exonuclease domain such P286R, V411L, L424V, S459F, P286H, F367S and L424I show a decrease in exonuclease activity, as measured by biochemistry experiments using purified Pole." There are no references for this statement (this is a review article, accurate references are essential) and it seems out of context, the authors should describe more about these variants in this section, or leave it to a later section but this one sentence alone seems incomplete, and needs more discussion. In addition, the section above this sentence is all about germline variants, however the variants listed in this sentence are found in somatic POLE mutations, this should be made clear.

Heterogeneity of the POLE mutation impact

3. Major Criticism: Again, not including references or citing the earlier work is a major weakness for this review, for example on page 7, the authors do not include all the references they can for many statements, "Besides the DNA mismatch repair defects that underlie Lynch syndrome, the mutations in POLE highlight the critical role of replication errors in predisposition to colorectal and endometrial cancers. This is in contrast to cancers of the breast and ovary, in which double-stranded DNA break repair is more significant in predisposition." There are many references to support this statement, however none are present. Review articles are a good source of finding important references, however this one does not include a large amount and misses that opportunity in multiple sections.

4. The title for this section implies they will discuss more details about the different variants, however this section does not do that. This section would benefit from a larger discussion about what is currently known about the variants. This entire section can be expanded to go into detail about each variant.

Mutator phenotype and mutation signature in POLE variants

5. Major Criticism: Again, this section falls short of including references, for example on page 8, this sentence is lacking

appropriate references only including later citations while ignoring earlier references (17, 59, 60) "a high increase in mutation rates have been documented in cancers with P286R, D275V, P286H, F367S, L424V, P436R, and S459F changes located closed to the DNA binding cleft of Pol (31,32)". And again, this sentence "Importantly, all of these variants lead to a much higher mutation frequency as compared to the variant that completely eliminates proofreading" is included without any references . Again, "Among hypermutated cancers presenting {greater than or equal to} 10 Mut/Mb, approximately 25% are associated with mutations in MMR genes alone, whereas a large number of ultra-hypermutated cancers ({greater than or equal to} 100 Mut/Mb) show mutations affecting both MMR and the replicative DNA polymerases, mainly POLE ", is missing citations. Again, on page 8, "Specific signature errors have been shown to be enriched in mutated POLE cancers. These include three trinucleotide hotspot mutations: C>A transversions in TCT context (C>A-TCT), C>T-TCG and T>G-TTT " should include references 17, 15, 59.

6. They should state what types of cancers have MMR and POLE mutations together, which types of cancer that does not occur, and in general expand this section to have a comprehensive review.

Mutated POLE tumors and immunogenic response

7. Major Criticism: This sentence mixes up very different types of high mutation load tumors, and consistent with other sections of this manuscript is missing citations. "It is becoming increasingly clear that cancers with high intrinsic mutation load (MMR and POLE mutated cancers) or cancers related to mutagenic environmental genotoxic exposure such lung, melanoma and bladder cancers, which show all a high mutational burdens, respond generally well to immunotherapy. " This entire section makes very bold statements with little to no references to back them up.

In this section, as in the rest of the manuscript, they do not make a clear distinction between somatic variants and germline. They discuss their N363K variant with no distinction that they are now describing a germline variant. There are recent papers examining POLE variants and response to therapeutics, in particular, (PMID: 34250404) is not included in this section, yet is a recent description of POLE variants and immune response.

In general, this review could benefit from a more complete survey of the current and past literature, adding relevant references, and expanding some sections as noted above.

Reviewer #2 (Comments to the Authors (Required)):

This is a well-written, concise review of the mechanisms and consequences of the different POLE mutations in cancer. The authors also propose some mechanistic basis underlining the mutation heterogeneity and discuss novel considerations for the choice and efficacy of therapies for POLE mutated tumors.

My only comment is not a mandatory request. I leave it to the authors to decide if they would like to add a short mention in the section "POLE mutation and cancer" of the recent study on single-strand events that could indicate how POLE mutations arise in cancer. They could refer to the publication of Gilad Evrony (Mei Hong Liu et al. 2023 bioRxiv doi: <https://doi.org/10.1101/2023.02.19.526140>).

Reviewer #3 (Comments to the Authors (Required)):

In review, the manuscript presents a summary of current literature on mammalian DNA polymerase involved in the replication of their leading and lagging strand. The authors have assembled and analyzed the current mutations in DNA polymerase epsilon and delta. In part they consider the association between mutant DNA polymerases and cancer.

The major conclusion of the paper is that mutations in these DNA polymerases are associated with specific types of human cancers. The data is well supported but does not necessarily indicate that the association is causative. The paper indicates that human tumors contain large numbers of different mutations. While it would be easy to suggest additional experiments, these would involve many laboratories and many years of research.

I suggest that the authors consider adding two columns to table one stating the locations of each of the mutations, and the second column indicating the enhancement of the mutation frequencies that have been reported. In addition, a clearer distinction between initiation and promotion of carcinogenesis appearing earlier in the paper may be useful. The legend to figure one contains a new original concept. The authors hypothesize that mutations in DNA polymerase could result in an enhanced mutagenesis without altering base specificity. In particular, studies on mutant polymerases may provide a new target for treatment options for specific cancers.

Response to reviewers' comments:**Black: Full reviewer' comments****Blue: our responses**

We are grateful to the reviewers for their constructive comments and suggestions on our review. Based on their comments, we changed critical parts throughout the manuscript. We have particularly included original citations and adequate references that were missing in the initial manuscript. We apologize for this weakness in the previous version of the paper and we agree that for a review, the citations are a critical component. The new parts of the text are highlighted in blue.

Please see below our detailed point by point response to the specific comments of each reviewer:

Reviewer #1:**Paragraph POLE mutations and cancer**1. Major Criticism:

This section starts with a description of MMR deficient cancers, then progresses into a description of POLE deficient cancers. The references for many sentences are incomplete or missing, here are some examples, (page 5) "POLE mutations have also been detected, albeit less frequently, in other types of gastrointestinal cancer, as well as in brain, breast, ovary, prostate, lung, kidney, cervix, and bone tumors (13,20-22). "does not include earlier papers that initially describe the variants, such as (17,59,60) and does reference databases such as COSMIC, cbiportal which is a bit confusing. Not including the appropriate references is a major problem throughout this manuscript.

We have now included in the entire paragraph appropriate references as required by the reviewer, especially for the types of cancers for which POLE has been found mutated (Campbell *et al.*, 2017; Cancer Genome Atlas, 2012; Cancer Genome Atlas Research *et al.*, 2013; Cerami *et al.*, 2012; Forbes *et al.*, 2015; Grossman *et al.*, 2016; Shinbrot *et al.*, 2014) (see page 5).

2. The end of this section (page6) contains this sentence, "Several POLE variants within the exonuclease domain such P286R, V411L, L424V, S459F, P286H, F367S and L424I show a decrease in exonuclease activity, as measured by biochemistry experiments using purified Polε." There are no references for this statement (this is a review article, accurate references are essential) and it seems out of context, the authors should describe more about these variants in this section, or leave it to a later section but this one sentence alone seems incomplete, and needs more discussion. In addition, the section above this sentence is all about germline variants, however the variants listed in this sentence are found in somatic POLE mutations, this should be made clear.

We apologize for the lack of references regarding the decreased in exonuclease activity of the POLE variants and the lack of clarity on germline vs somatic variants. We have now introduced original references showing that these POLE mutations affect the activity of the exonuclease (Korona *et al.*, 2011; Parkash *et al.*, 2019; Shinbrot *et al.*, 2014) (see page 6).

We have also better described the germline versus somatic POLE mutations : see page 5 the paragraph " The last decade has witnessed the identification in cancers from many tissue types of multiple somatically acquired missense mutations clustering in the sequence encoding the exonuclease proofreading domain of POLE....." and page 5-6 the paragraph "Mutations in the

exonuclease domain of POLE can also be inherited through the germline, leading to a rare autosomal dominant familial cancer predisposition syndrome documented as polymerase proofreading-associated polyposis (PPAP), characterized.....”

Paragraph Heterogeneity of the POLE mutation impact

3. Major Criticism: Again, not including references or citing the earlier work is a major weakness for this review, for example on page 7, the authors do not include all the references they can for many statements, "Besides the DNA mismatch repair defects that underlie Lynch syndrome, the mutations in POLE highlight the critical role of replication errors in predisposition to colorectal and endometrial cancers. This is in contrast to cancers of the breast and ovary, in which double-stranded DNA break repair is more significant in predisposition." There are many references to support this statement, however none are present. Review articles are a good source of finding important references, however this one does not include a large amount and misses that opportunity in multiple sections.

We have also added in this paragraph some aspects on the comparison between POLE and POLD1 mutations in hypermutated cancers in order to make the review more robust (see page 7 the paragraph : “Generally, there are much less cancer driver mutations in POLD1 than in POLE in human cancers. This might be due to the reduced fitness and viability of POLD1 mutants as Pol δ holds multiple critical roles besides lagging strand replication, including its ability to proofread in trans the errors made by Pol ϵ and Pol α , its role during MMR and during Okazaki fragment maturation”).

4. The title for this section implies they will discuss more details about the different variants, however this section does not do that. This section would benefit from a larger discussion about what is currently known about the variants. This entire section can be expanded to go into detail about each variant.

The reviewer is correct and we apologize. As required by the reviewer, we have now expanded the paragraph page 8 by describing more precisely the differential mutagenic impact of the POLE variants in haploid and diploid yeast, and the lack of correlation between the increase of mutation rate, the TMB and the frequency of the variant in tumors.

Paragraph Mutator phenotype and mutation signature in POLE variants

5. Major Criticism:

Again, this section falls short of including references, for example on page 8, this sentence is lacking appropriate references only including later citations while ignoring earlier references (17, 59, 60) "a high increase in mutation rates have been documented in cancers with P286R, D275V, P286H, F367S, L424V, P436R, and S459F changes located closed to the DNA binding cleft of Pol ϵ (31,32)"

As required, we have now included earlier references to the section

and again, this sentence "Importantly, all of these variants lead to a much higher mutation frequency as compared to the variant that completely eliminates proofreading" is included without any references .

We apologize, we have now incorporated the paper by Kane and Sherbakova in 2014, which described that cancer-associated POE variants can produce unusually strong mutator phenotype exceeding that of proofreading-deficient mutants by up to two orders of magnitude.

Again, "Among hypermutated cancers presenting {greater than or equal to} 10 Mut/Mb, approximately 25% are associated with mutations in MMR genes alone, whereas a large number of ultra-hypermutated cancers ({greater than or equal to} 100 Mut/Mb) show mutations affecting both MMR and the replicative DNA polymerases, mainly POLE ", is missing citations.

We have now incorporated the appropriate reference by Jansen et al. in 2016.

Again, on page 8, "Specific signature errors have been shown to be enriched in mutated POLE cancers. These include three trinucleotide hotspot mutations: C>A transversions in TCT context (C>A-TCT), C>T-TCG and T>G-TTT "should include references 17, 15, 59.

We have now incorporated these references

6. They should state what types of cancers have MMR and POLE mutations together, which types of cancer that does not occur, and in general expand this section to have a comprehensive review.

We have now described that the combined MMR and POLE defect were observed in ultra-mutated Lynch syndrome tumors (Jansen et al., 2016), endometrial and colorectal cancers (Shlien et al., 2015), as well as glioblastoma (Shlien et al. 2015).

Paragraph Mutated POLE tumors and immunogenic response

7. Major Criticism:

This sentence mixes up very different types of high mutation load tumors, and consistent with other sections of this manuscript is missing citations. "It is becoming increasingly clear that cancers with high intrinsic mutation load (MMR and POLE mutated cancers) or cancers related to mutagenic environmental genotoxic exposure such lung, melanoma and bladder cancers, which show all a high mutational burdens, respond generally well to immunotherapy. " This entire section makes very bold statements with little to no references to back them up.

Again, we apologize and we have now incorporated the appropriate reference (Le DT 2015, Le DT 2017, (McGranahan et al, 2016; Rizvi et al, 2015; Yarchoan et al, 2017)

In this section, as in the rest of the manuscript, they do not make a clear distinction between somatic variants and germline. They discuss their N363K variant with no distinction that they are now describing a germline variant.

We have already modified page 5 for this issue and better described the germline versus somatic POLE mutations.

The effect of POLE variant in term of chromosome instability and DNA damage and the associated mechanisms is effective when it is expressed, as somatic or germline, so this is the reason we discussed the general mechanistic aspects independently of the type of the variant.

There are recent papers examining POLE variants and response to therapeutics, in particular, (PMID: 34250404) is not included in this section, yet is a recent description of POLE variants and immune response.

We thank the reviewer for this reference that we have now added page 11 (Keshinro et al., JCO Precis Oncol 2021)

Reviewer #2:

We thank this reviewer for his/her global positive appreciation of our review.

We thank also the reviewer for his/her suggestion to add in the section "POLE mutation and cancer" the recent study on single-strand events that could indicate how POLE mutations arise in cancer and refer to the publication of Gilad Evrony (Mei Hong Liu et al. 2023 bioRxiv). This work, which profiled samples from individuals with cancer-predisposition syndromes and defined single-strand mismatch signatures, shows correspondences between single-strand signatures and known double-strand mutational signatures induced by defective proofreading, so this a possible mechanism of the generation of mutation signature in POLE mutants but we have decided not to include it since this mechanistic aspect is not developed in our review, and would require the description of additional alternative mechanisms .

Reviewer #3:

The reviewer proposes to incorporate to table 1 the locations of each of mutation and the corresponding enhancement of the mutation frequencies that have been reported. We thank the reviewer for this suggestion. We have corrected the table 1 with the required items.

The reviewer proposes also to include a clearer distinction between initiation and promotion of carcinogenesis.

We have better described such distinction in page 5 : "Such a defective proofreading activity producing a mutator phenotype, which have been established in model systems, such as yeast, bacteria, and mice, lead to tumorigenesis. These *POLE* variants are present in heterozygous tumors with no apparent loss of heterozygosity (LOH) and with high mutation loads, up to 500 mutations per megabase (Mut/Mb). A strong mutator phenotype in the presence of the wild-type allele is consistent with the participation of both the wild-type and the mutant polymerases in DNA replication, in contrast to mutated MMR tumors, where loss of both alleles is required to produce a mutator effect. It has been proposed that differential expression levels of the wild-type and mutant *POLE* alleles in the course of cancer progression may allow transient stages of hypermutation that promote tumor growth together with a threshold limiting excessive mutation load to maintain fitness.

Finally, we thank the reviewer who commented on a new original concept in Figure 1 and the consequences for new treatment options for specific cancers.

October 2, 2023

RE: Life Science Alliance Manuscript #LSA-2023-02290-TR

Dr. Jean-Sébastien Hoffmann
University Cancer Institute Toulouse Oncopole
IUCT ONCOPOLE
1, avenue Irène-Joliot-Curie
Toulouse 31059
France

Dear Dr. Hoffmann,

Thank you for submitting your revised manuscript entitled "Exploring the basis of heterogeneity of cancer aggressiveness among the mutated POLE variants". We would be happy to publish your paper in Life Science Alliance pending final revisions necessary to meet our formatting guidelines.

- please upload your figure as a single file, and remove it from the main manuscript text
- please add the Twitter handle of your host institute/organization as well as your own or/and one of the authors in our system
- please add ORCID ID for the secondary corresponding author--they should have received instructions on how to do so
- please add an Author Contributions section to your main manuscript text
- please add your figure and table legends to the main manuscript text after the references section
- please add a conflict of interest statement to your main manuscript text
- please add a callout for Figure 1A to your main manuscript text

A. FINAL FILES:

B. MANUSCRIPT ORGANIZATION AND FORMATTING:

Sincerely,

October 18, 2023

RE: Life Science Alliance Manuscript #LSA-2023-02290-TRR

Dr. Jean-Sébastien Hoffmann
University Cancer Institute Toulouse Oncopole
IUCT ONCOPOLE
1, avenue Irène-Joliot-Curie
Toulouse 31059
France

Dear Dr. Hoffmann,

Thank you for submitting your Review entitled "Exploring the basis of heterogeneity of cancer aggressiveness among the mutated POLE variants". It is a pleasure to let you know that your manuscript is now accepted for publication in Life Science Alliance. Congratulations on this interesting work.

DISTRIBUTION OF MATERIALS:

Again, congratulations on a very nice paper. I hope you found the review process to be constructive and are pleased with how the manuscript was handled editorially. We look forward to future exciting submissions from your lab.

Sincerely,
